Harnessing the power of comparative genomics to support the distinction of sister species within Phyllosticta and development of highly specific detection of Phyllosticta citricarpa causing citrus black spot by real-time PCR

http://orcid.org/0000-0001-9359-5098 Ioos Renaud 1 renaud.ioos@anses.fr
Puertolas Alexandra 1 2
Renault Camille 1 2
Ndiaye Aida 1
Cerf-Wendling Isabelle 1
Hubert Jacqueline 1
Wang Wen 3
Jiao Chen 3
Li Hongye 3
Armengol Josep 4
Aguayo Jaime 1
1 Laboratoire de la Santé des Végétaux, Unité de Mycologie, USC INRAE, ANSES , Malzéville , France
2 ANSES, European Union Reference Laboratory on Plant Pathogenic Fungi and Oomycetes , Malzéville , France
3 The Key Laboratory of Biology of Crop Pathogens and Insects of Zhejiang Province, Institute of Biotechnology, Zhejiang University , Zhejiang, Hangzhou , China
4 Instituto Agroforestal Mediterráneo, Universitat Politècnica de València , Valencia , Spain
Shamoun Simon
Electronic publication date: 2023 Oct 23
Publication date: 2023
Volume: 11
Electronic Location ID: e16354
Received 2023 May 4; Accepted 2023 Oct 4
Copyright: © 2023 Ioos et al.
Copyright year: 2023
Copyright holder: Ioos et al.
License: This is an open access article distributed under the terms of the Creative Commons Attribution License, which permits unrestricted use, distribution, reproduction and adaptation in any medium and for any purpose provided that it is properly attributed. For attribution, the original author(s), title, publication source (PeerJ) and either DOI or URL of the article must be cited.
License URL: https://creativecommons.org/licenses/by/4.0/

Keywords: Detection, Plant disease, Quarantine, Validation, Comparative genomics, Fungus

Funding: ANSES Plant Health Laboratory (LSV) French National Research Agency (ANR) French government’s “Investing for the Future” (PIA) program ANR-11-LABX-0002-01 EU Grant S12.809457 European Union Reference Laboratory’s Work Program on Fungi Pathogenic to Plants National Natural Science Foundation of China 32102148 China Agriculture Research System CARS-26 The mycology research unit of the ANSES Plant Health Laboratory (LSV) is supported by a grant managed by the French National Research Agency (ANR) as part of the French government’s “Investing for the Future” (PIA) program (ANR-11-LABX-0002-01, Laboratory of Excellence-ARBRE). The specific work focusing on the improvement and validation of the CBS detection tool was supported by EU Grant S12.809457, in the framework of the European Union Reference Laboratory’s work program on fungi pathogenic to plants. The team at Zhejiang University was supported by the National Natural Science Foundation of China (No: 32102148) and China Agriculture Research System (CARS-26). The funders had no role in study design, data collection and analysis, decision to publish, or preparation of the manuscript.

==============================
Citrus crops are affected by many fungal diseases. Among them, Citrus Black Spot caused by the ascomycete Phyllosticta citricarpa is particularly economically damaging wherever it occurs. Many other species of Phyllosticta are described on Citrus, but only P. citricarpa is considered a quarantine pest on the European continent. In order to prevent the introduction of this species into Europe, it is essential to have a detection test which can reliably identify it, and not confuse it with other species present on citrus, notably P. paracitricarpa. The latter taxon has recently been described as very close to P. citricarpa, and most detection tests do not allow to distinguish the two species. In this work, we exploited the genomic data of 37 isolates of Phyllosticta spp. from citrus, firstly to assess their phylogenetic relationships, and secondly to search for genomic regions that allowed the definition of species-specific markers of P. citricarpa. Analysis of 51 concatenated genes separated P. citricarpa and P. paracitricarpa in two phylogenetic clades. A locus was selected to define a hydrolysis probe and primers combination that could be used in real-time PCR for the specific detection of the quarantine species, to the exclusion of all others present on Citrus. This test was then thoroughly validated on a set of strains covering a wide geographical diversity, and on numerous biological samples to demonstrate its reliability for regulatory control. The validation data highlighted the need to check the reliability of the test in advance, when a change of reagents was being considered.

Introduction

The genus Phyllosticta comprises dozens of species with different ecological behaviors; some are endophytes, others are plant pathogens and a few are even saprobes (Wikee et al., 2013). Pathogenic species of Phyllosticta affect a broad range of hosts, and are responsible for numerous diseases, including leaf and fruit spots. On citrus, three species—namely P. capitalensis, P. citribraziliensis, and P. paracapitalensis—are described as endophytes (Glienke et al., 2011; Guarnaccia et al., 2019), whereas five species are described as pathogens. P. citricarpa causes citrus black spot (CBS) (Kiely, 1948; Van der Aa, 1973), P. citriasiana and P. citrimaxima both cause citrus tan spot of Citrus maxima (Wikee et al., 2013; Wulandari et al., 2009), and P. citrichinaensis causes spots and freckle on several Citrus species (Wang et al., 2012). Recently, Guarnaccia et al. (2017) described a new species, Phyllosticta paracitricarpa, which is genetically closely related to P. citricarpa. P. paracitricarpa was isolated from Citrus limon leaf litter in Greek lemon orchards (Guarnaccia et al., 2017) as well as from lemon fruits in China (Wang et al., 2023) and has been shown to be pathogenic to C. sinensis and C. limon after artificial inoculations on detached leaves and fruits (Guarnaccia et al., 2017; Wang et al., 2023). It was also retrospectively demonstrated that some Chinese isolates causing black spots on mandarin fruits, tentatively named “P. citricarpa subclade II” (Wang et al., 2012) could be re-assigned to P. paracitricarpa according to a multilocus phylogenetic analysis using six markers (Guarnaccia et al., 2019). The description of this new sister species was based on a few fixed nucleotide polymorphisms within one gene from the core genome (translation elongation factor 1-alpha gene, tef1) and within a region of the rDNA operon (Large Subunit, LSU), combined with some morphological differences with P. citricarpa. In general, the morphological features of P. paracitricarpa in pure culture are very similar to P. citricarpa and it is virtually impossible to differentiate the two species based solely on microscopic traits (EPPO, 2020).

CBS is an important disease outside Europe that causes significant economic losses in orchards. This disease is named after the characteristic symptoms of citrus fruits, i.e., black spots on the citrus peel, although the symptomatology of this disease can be very variable, ranging from hard spots, virulent spots or false melanose spots to freckle spots. This variability in symptoms is the result of the temperature and maturity of the affected fruits (EPPO, 2020). Phyllosticta citricarpa has a wide host range and is able to infect mostly all Citrus spp. (except C. aurantium and hybrids, and C. latifolia), Fortunella spp. and Poncirus spp. Although present in many citrus growing regions, some regions are still considered as disease-free (EPPO, 2022). In these regions (or countries) P. citricarpa is generally listed as a quarantine organism (EPPO, 2022; European Union Commission, 2019). According to the European Food Safety Agency (EFSA, 2014), the main pathway for dispersal and introduction of P. citricarpa is through the trade of citrus fruits and plants for planting (EFSA, 2014). For instance, if this pathogen was to be introduced into the European Union (EU), it would be able to become established due to the presence of hosts and favorable climatic conditions for its spread and disease development, as was the case recently in Tunisia (Boughalleb-M’Hamdi et al., 2020; Galvañ et al., 2022). While the importation of citrus plants for planting is prohibited by current EU regulations (European Union Commission, 2019), the importation of fruit shipments for consumption is frequent, providing that they are free from P. citricarpa. Despite its pathogenicity toward citrus fruits, the sister species P. paracitricarpa is currently not considered a quarantine pathogen in any region of the world. Therefore, it is necessary to test fruits for the presence of P. citricarpa and a highly specific detection assay enabling a quick response from the analysis lab is required so that citrus shipments may be blocked or released within only a few days.

The current diagnostic protocols for detecting and identifying P. citricarpa include several real-time PCR (qPCR) methods (Ahmed et al., 2020; Schirmacher et al., 2019; van Gent-Pelzer et al., 2007; Zajc et al., 2022) and conventional PCR (cPCR) methods (Baayen et al., 2002; Peres et al., 2007). Most of these assays are based on the amplification of a specific region of the internal transcribed spacer (ITS) region of rDNA (Bonants et al., 2003; Peres et al., 2007; Schirmacher et al., 2019; van Gent-Pelzer et al., 2007). However, it has been shown that the ITS is 100% identical in the two sister species P. citricarpa and P. paracitricarpa (Guarnaccia et al., 2017). False positive results can therefore be obtained when testing DNA of the non-target species P. paracitricarpa (EPPO, 2020). Such cross-reactions with P. paracitricarpa DNA were also observed with the assay developed by Ahmed et al. (2020) targeting phylogenetic marker MCM7. The method of Zajc et al. (2022) is based on the tef1 gene, which should differ between P. citricarpa and P. paracitricarpa by five nucleotide changes (Guarnaccia et al., 2017). However, this method was developed and validated in vitro using two strains of P. paracitricarpa, both from the same orchard in Greece. This may be an additional difficulty when designing specific PCR methods targeting quarantine organisms. For example, it has been shown that P. citricarpa exhibits intraspecific polymorphism in some relevant genes frequently used in fungal phylogenetics and taxonomy, such as γ-actin (actA), the glyceraldehyde-3-phosphate dehydrogenase (gadph), or the LSU (Guarnaccia et al., 2017; Wang et al., 2012).

It is a challenge to find diverging regions within the genomes of genetically closely related or cryptic species (Dutech et al., 2016; Feau et al., 2018). For example, according to Guarnaccia et al. (2019), a very limited proportion of genes appeared to differ between P. citricarpa and P. paracitricarpa, although a limited number of strains of P. paracitricarpa were included in the study. Indeed, for those particular taxa, frequently used housekeeping genes are not sufficiently polymorphic to be used as targets for highly specific PCR or real-time PCR oligonucleotides. In the case of a species complex, genetic lineages of the same species, or cryptic species, comparative genomics is a powerful means for screening either polymorphic or unique regions in the genomes (Bergeron et al., 2019; Feau et al., 2019; Pieck et al., 2017; Thierry et al., 2019).

The first objective of this study was to use a set of genomes from P. citricarpa, P. paracitricarpa, and other Phyllosticta species that are pathogenic or endophytic in Citrus so as to assess their phylogenetic relationship. Second, we took advantage of this comparative genomics study to screen polymorphic genomic regions, which could be used as tentative markers to design a specific PCR assay targeting P. citricarpa. Last, a set of diagnostic tools (real-time and conventional PCR) were designed and fully validated for the specific in planta detection of P. citricarpa, allowing it to be differentiated from its closely related sister species, P. paracitricarpa, and from other Phyllosticta species described in Citrus. In this work, a large set of Phyllosticta strains from Citrus were used as references, both for sequencing and for testing, in order to cover insofar as possible the range of natural diversity occurring for these globally distributed species.

Material and Methods

Genomics and phylogenetics of Phyllosticta spp. in Citrus

The genomes of 14 strains representing P. citricarpa (six), P. paracitricarpa (two), P. citriasiana (three), P. capitalensis (one), P. citrichinaensis (one), and Phyllosticta sp. (one) were sequenced (Information S1). In order to take into consideration potential intraspecific diversity within P. citricarpa, strains from different regions of the world were included: Togo, South Africa, Zimbabwe, Tunisia and Malta. Genomic DNA (gDNA) was extracted from monosporic cultures using the GenElute Plant Genomic DNA Miniprep kit (Merck, Lebanon, NJ, USA), following the manufacturer’s instructions, after an initial grinding step using lysing Matrix C tubes in a FastPrep24 homogenizer (MP Biomedicals, Santa Ana, CA, USA), with a one-run program at 6.0 U for 1 min. DNA samples were eluted twice in a volume of 50 µL each time. gDNA concentrations were estimated using a Qubit fluorometer (Thermo Fisher Scientific, Illkirch-Graffenstaden, France) prior to genome sequencing. Library construction and paired-end genome sequencing (2 × 150 bp) were performed by GENEWIZ (Azenta Life Sciences, Leipzig, Germany) using an Illumina HiSeq device. Additional genomes from P. citricarpa (11), P. paracitricarpa (three), and P. citribraziliensis (one) were retrieved either from the Joint Genome Institute Genome Portal (https://genome.jgi.doe.gov/portal/) or the National Center for Biotechnology Information (https://www.ncbi.nlm.nih.gov/genome/) (Information S1).

TrimGalore-0.6.8 (https://www.bioinformatics.babraham.ac.uk/projects/trim_galore/), a wrapper tool around Cutadapt (Martin, 2011) and FastQC (https://www.bioinformatics.babraham.ac.uk/projects/fastqc/) was used for adapter trimming and the quality control of genomic reads. De novo genome assemblies were conducted with ABySS-2.3.1 (Jackman et al., 2017; Simpson et al., 2009). Assembly quality statistics were obtained with Quast 5.0.2 (Gurevich et al., 2013) and BUSCO 5.2.2 (Manni et al., 2021). Gene prediction was performed on assembled genomes using Augustus 3.4.0 (Stanke et al., 2006). Single-copy genes were identified using Funybase (Marthey et al., 2008), a reliable database that provides 246 orthologous gene families for performing comparative and phylogenetic analyses in fungi. These gene clusters are present as single copies in 21 fungal genomes. The protein sequences of Sclerotinia sclerotiorum were searched for in the annotated genomes of P. citricarpa using BLASTp with a similarity cutoff e-value of 1−20 (Feau et al., 2018). Sequences with more than one hit in the target genome were discarded from subsequent analyses. Single-copy isolated protein sequences were then searched for in the annotated genomes in order to obtain the nucleotide sequences (predicted genes) obtained with Augustus. The isolates retained for this part of the study were those in which all the selected genes were present and not fragmented or missing. Genome assemblies were deposited at DDBJ/ENA/GenBank under the BioProject number PRJNA949004.

These nucleotide sequences were individually analyzed using SeaView version 5 (Gouy et al., 2021). The analyses included sequence alignment with Muscle (Edgar, 2004), removal of poorly aligned positions with Gblocks (Talavera & Castresana, 2007) and concatenation of all the isolated single-copy sequences. RAxML (Stamatakis, 2014) was used for maximum-likelihood phylogenetic inference, using the general time reversible (GTR)-gamma model with 1,000 replicates to estimate bootstrap supports. A Bayesian phylogenetic tree was inferred using MrBayes 3.2 (Ronquist et al., 2012). Runs were performed using the Bayesian MCMC model jumping approach. Four MCMC chains were run using the default heating with tree sampling performed every 5,000 generations. Runs were performed for at least 10 million generations, and stopped when the standard deviation of split frequencies was below 0.01. A consensus maximum-likelihood and Bayesian phylogenetic cladogram was visualized and annotated with SplitsTreeCE (https://github.com/husonlab/splitstree6) (Huson & Cetinkaya, 2023).

Selection of highly divergent genomic regions in P. citricarpa and P. paracitricarpa

The genomes of five P. paracitricarpa and 32 P. citricarpa strains were used to check which regions in the two species were clearly divergent (Information S1). Genomic resources included the genomes of strains sequenced in this study and 30 previously published genomes (Coetzee et al., 2021; Guarnaccia et al., 2019; Rodrigues et al., 2019) which were downloaded either from the MycoCosm database or the National Center of Biotechnology Information (NCBI) Sequence Read Archive (SRA) database.

Both the newly generated reads and the downloaded sequences were individually filtered by Trimmomatic v0.39 (Bolger, Lohse & Usadel, 2014) to remove adapters and low-quality sequences. The cleaned reads were then mapped to the reference genome CBS 102373 using BWA-mem v0.7.17 (Li & Durbin, 2009). The resulting alignments were sorted by position using SAMtools v1.3 (Li et al., 2009) and the duplicated alignments from the PCR amplification were then masked by Picard v2.18.7 (http://broadinstitute.github.io/picard/). The Genome Analysis Toolkit (GATK) v4.2 (McKenna et al., 2010) was used for variant calling, selection, and filtration. The obtained variants were filtered using the following parameters: “QD < 2.0 || MQ < 40.0 || FS > 60.0 || SOR > 3.0 || MQRankSum < −12.5 || ReadPosRankSum < −8.0”. Polymorphic sites with a missing rate less than 0.5 and a minor allele frequency greater than 0.02 were kept.

Regions that were clearly distinct from each other in the two Phyllosticta species were identified by comparing the differences in allele frequency in the two populations; the read alignments were visualized using the Integrative Genomics Viewer (IVG) (Robinson et al., 2017). Read alignments of the ITS and tef1 genomic regions in the reference genome were also checked in order to assess the intra- and interspecific polymorphism within these genes recommended as a barcode for identification purposes (EPPO, 2020; Schoch et al., 2012). The genomic positions of the ITS and tef1 gene were identified by searching for the target regions of the following PCR primers: V9G (de Hoog & van den Ende, 1998), ITS4 (White et al., 1990), EF1-728F (Carbone & Kohn, 1999), and EF-2 (O’Donnell et al., 1998). The reads in the candidate regions were extracted from the bam file using SAMtools, and visualized with the IGV.

Design of P. citricarpa-specific primers and probe for PCR and real-time PCR

The previous steps led to the identification of a genomic region that was clearly different in P. citricarpa and P. paracitricarpa. This region of unknown function was downloaded for each genome for P. citricarpa, P. paracitricarpa, and other Phyllosticta species occurring in Citrus. All the orthologous sequences were aligned using the MUSCLE algorithm implemented in Geneious Prime (Biomatters V2021.2.2; Biomatters, Auckland, New Zealand). Candidate primers and probe oligonucleotides were designed using Primer 3 (Untergasser et al., 2012) based on the regions showing polymorphisms between species, but conserved in P. citricarpa. The melting temperature, potential formation of secondary structures, and interactions among the oligonucleotide sequences were evaluated in silico using Geneious Prime. All the primers and probes used in this study were synthesized by Eurogentec (Seraing, Belgium).

Fungal strains and extraction of genomic DNA from single-spore cultures

A panel of 56 Phyllosticta spp. strains from different origins and hosts (Table 1), mainly from Citrus spp., were used for this study. All these isolates were identified by barcode sequencing using either the regions LSU and tef1 (for Phyllosticta spp.) or ITS (other species) (Table 2). A subset of sequence data was submitted to GenBank (Information S2). In addition, a subset of the P. citricarpa and P. paracitricarpa strains were examined by genotyping using ten microsatellite loci, initially developed for P. citricarpa, according to Carstens et al. (2017) and Wang et al. (2016). A minimum spanning network was constructed based on pairwise allele shared distance, using Edenetwork 2.18 (Kivelä, Arnaud-Haond & Saramäki, 2015).

Table 1 Isolates of fungal species used in this study.

Species	Isolate	Host	Country or region of origin	Year	qCBS-qPCR-F/R/P	cCBS-cPCR-F/R	ITS PCR	
Phyllosticta citricarpa	LSVM 205	Citrus sp.	South Africa	2009	30.28 ± 0.07	+	+	
	LSVM 359	Citrus sp.	Togo	2010	19.87 ± 0.02	+	+	
	LSVM 494	Citrus sp.	Togo	2011	30.91 ± 0.14	+	+	
	LSVM 1101	C. sinensis × P.trifoliata	South Africa	2014	20.56 ± 0.03	+	+	
	LSVM 1102	C. sinensis	South Africa	2013	20.59 ± 0.26	+	+	
	LSVM 1103	C. sinensis	South Africa	2013	31.45 ± 0.38	+	+	
	LSVM 1116	C. sinensis	South Africa	2014	19.82 ± 0.00	+	+	
	LSVM 1123	C. sinensis	Zimbabwe	2014	21.06 ± 0.00	+	+	
	LSVM 1228	C. sinensis	Ivory Coast	2015	20.59 ± 0.07	+	+	
	LSVM 1243	C. sinensis	South Africa	2015	20.41 ± 0.05	+	+	
	LSVM 1244	C. sinensis	Brazil	2015	20.70 ± 0.18	+	+	
	LSVM 1266	C. sinensis	Brazil	2015	37.31 ± 0.22	+/−	+	
	LSVM 1268	C. sinensis	Brazil	2015	19.93 ± 0.11	+	+	
	LSVM 1270	C. sinensis	Brazil	2015	21.74 ± 0.15	+	+	
	LSVM 1276	C. sinensis	Brazil	2015	30.82 ± 0.35	+	+	
	CBS 141350	C. sinensis (litter)	Malta	2016	21.51 ± 0.05	+	+	
	CBS 141351	C. sinensis (litter)	Portugal	2016	23.44 ± 0.09	+	+	
	LSVM 1499	C. limon	Tunisia	2019	16.30 ± 0.08	+	+	
	LSVM 1500	C. limon	Tunisia	2019	20.04 ± 0.47	+	+	
	LSVM 1501	C. sinensis	Tunisia	2019	20.56 ± 0.10	+	+	
	LSVM 1502	C. sinensis	Tunisia	2019	20.28 ± 0.01	+	+	
	GC-115	C. sinensis	Brazil	2010	23.28 ± 0.20	+	+	
	GC-129	C. limon	Argentina	2014	23.15 ± 0.26	+	+	
	GC-130	C. limon	Argentina	2014	23.70 ± 0.00	+	+	
	GC-131	C. limon	Argentina	2014	23.04 ± 0.24	+	+	
	GC-133	C. sinensis	Angola	2016	24.31 ± 0.01	+	+	
P. paracitricarpa	CBS141357	C. lemon (litter)	Greece	2016	>45	–	+	
	CBS 141359	C. lemon (litter)	Greece	2016	>45	–	+	
	ZJUCC200937	C. reticulata	China	2017	>45	–	+	
	GIHF 303	C. sinensis	China	2017	>45	–	+	
	LSVM 1238*	C. pennivesiculata	Bengladesh	2015	>45	–	+	
P. citriasiana	LSVM 1146	C. maxima	China	2014	>45	–	+	
	LSVM 204	Citrus sp.	–	2007	>45	–	+	
	LSVM 608	C. maxima	China	2012	>45	–	+	
	LSVM 903	C. maxima	China	2013	>45	–	+	
	CBS 123393	Citrus sp.	Vietnam	2014	>45	–	+	
	LSVM 1147	C. maxima	China	2014	>45	–	+	
	LSVM 1152	C. maxima	China	2014	>45	–	+	
	LSVM 1165	C. maxima	China	2014	>45	–	+	
	LSVM 1174	C. paradisi	USA	2014	>45	–	+	
	LSVM 1279	C. maxima	China	2015	>45	–	+	
P. capitalensis	LSVM 502	Musa sp.	Guadeloupe	2012	>45	–	+	
	LSVM 607	C. maxima	China	2012	>45	–	+	
	LSVM 1089	C. latifolia	Brazil	2014	>45	–	+	
	LSVM 1104	C. sinensis × C. trifoliata	Brazil	2014	>45	–	+	
	LSVM 1117	C. latifolia	Guadeloupe	2014	>45	–	+	
	LSVM 1119	C. paradisi	Mayotte	2014	>45	–	+	
	LSVM 1124	C. paradisi	Mayotte	2014	>45	–	+	
	LSVM 1163	C. grandis	China	2014	>45	–	+	
	LSVM 1166	C. grandis	China	2014	>45	–	+	
	LSVM 1226	Rollinia pulchrinervia	French Guyana	2015	>45	–	+	
	LSVM 1245	C. sinensis	Brazil	2015	>45	–	+	
P. citrichinaensis	CBS 129764	C. maxima	China	2015	>45		+	
P. citribraziliensis	CBS 100098	Citrus sp.	Brazil	2014	>45	–	+	
P. paracapitalensis	CBS 141353	C. floridana	Italy	2016	>45	–	+	
Phyllosticta sp.	LSVM 1173	C. latifolia	Mexico	2014	>45	–	+	
Colletotrichum gloeosporioides	LSVM 935	C. sinensis	Guadeloupe	2013	>45	–	+	
Fusarium oxysporum	LSVM 902	C. sinensis	Guadeloupe	2013	>45	–	+	
Rhizoctonia solani	LSVM 392	Platanus sp.	Corse	2011	>45	–	+	
Passalora loranthi	LSVM 1133	C. sinensis	Brazil	2014	>45	–	+	
Elsinoe fawcettii	DUDA IMK TEM 3	Citrus (Temple fruit)	Florida	2001	>45	–	+	
Elsinoe fawcettii	CBS233.64	C. aurentium	Panama	–	>45	–	+/−	
Elsinoe fawcettii	USA Russel 15	C. temple	Florida	1991	>45	–	+	
Elsinoe fawcettii	CC-3	Citrus volkamer	Florida	1990	>45	–	+	
Elsinoe fawcettii	PK MURL3	Citrus murcott	Florida	2014	>45	–	+	
Elsinoe fawcettii	DUDA IMK TEM 2	Citrus sp.	–	–	>45	–	–	
Elsinoe australis	CBS22964	C. aurantifolia	Brazil	–	>45	–	+	
Elsinoe australis	SOS 53525	Sweet orange	Florida	2016	>45	–	+	
Elsinoe citricola	CBS 141876	C. limonia	Brazil	2010	>45	–	–	
Alternaria alternata	LSVM 1108	C. latifolia	Brazil	2012	>45	–	NT	
Pseudocercospora angolensis	C-ethiop-3	C. aurantiifolia	Ethiopia	2016	>45	–	+	
Diaporthe sp.	LSVM 1109	C. latifolia	Brazil	2012	>45	–	+	
Fusarium proliferatum	LSVM 1168	C. sinensis	Argentina	2014	>45	–	+	
Neofusicoccum parvum	LSVM 884	C. sinensis	Guadeloupe	2013	>45	–	+	
Periconia sp.	LSVM 1139	C. sinensis	Argentina	2014	>45	–	+	
Penicillium italicum	Penicillium 1	Citrus sp.		x	>45	–	+	
Ramichloridium cerophilum	LSVM 1176	C. grandis	China	2014	>45	–	+	
Notes:

DNA template concentrations were all adjusted to 1 ng/µL for PCR testing. NT, not tested.

* Strain LSVM1238 was initially identified as P. citricarpa by sequencing EF1 and ITS according to Guarnaccia et al. (2017), but displays a microsatellite multilocus genotype pattern typical of P. paracitricarpa. It also clusters with P. paracitricarpa strains in the 51-gene phylogeny.

Table 2 List of primers and probes used in this study for conventional and real-time PCR.

Target (DNA region)	PCR test	Purpose	Name	Sequence (5′–3′)	Reference	
Phyllosticta citricarpa (Scaffold_23 CBS102373)	“cCBS” Conventional PCR	Detection of P. citricarpa	CBS-F	TCCTTTGGAGCAGCTGC	This study	
	CBS-R	CTTGCTTCCCTTGAATGAGACTG		
“qCBS” Real-time PCR	Detection of P. citricarpa	CBS-F	TCCTTTGGAGCAGCTGC		
	CBS-R	CTTGCTTCCCTTGAATGAGACTG	This study	
	CBS-P	FAM-AGTCACCTCCGAAGAAGCCAGTCC-BHQ1		
Phyllosticta citricarpa (ITS)	“VGP” Real-time PCR	Detection of P. citricarpa	GcF1	GGTGATGGAAGGGAGGCCT	van Gent-Pelzer et al. (2007)	
GcR1	GCAACATGGTAGATACACAAGGGT		
GcP1	FAM-AAAAAGCCGCCCGACCTACCTTCA-BHQ1		
Plant/fungus (18S rDNA)	Real-time PCR	DNA quality control	18S uni F	GCAAGGCTGAAACTTAAAGGAA	Ioos et al. (2009)	
	18S uni R	CCACCACCCATAGAATCAAGA		
	18S uni P	JOE-ACGGAAGGGCACCACCAGGAGT-BHQ1		
ITS region	Conventional PCR	Sequencing	ITS5	GGAAGTAAAAGTCGTAACAAGG	White et al. (1990)	
		ITS4	TCCTCCGCTTATTGATATGC		
tef1 gene	Conventional PCR	Sequencing	EF1-728F	CATCGAGAAGTTCGAGAAGG	Carbone & Kohn (1999)	
		EF2	GGARGTACCAGTSATCATGTT	O’Donnell et al. (1998)	
28 S large subunit rDNA	Conventional PCR	Sequencing	LR0R	ACCCGCTGAACTTAAGC	Moncalvo (1995)	
LR5	TCCTGAGGGAAACTTCG	Vilgalys & Hester (1990)	

P. citricarpa strains were manipulated in a biosafety level 3 laboratory. All the cultures were single-spored, and then grown at 22 °C on potato dextrose agar (PDA) with a sterile cellophane disc over the surface. Once the mycelium colonized the cellophane disc, it was recovered with a sterile scalpel and placed in a sterile microtube, then kept at −20 °C until DNA extraction.

Additional strains of other genera occurring on Citrus were included in this study (Table 1). Cultures were grown as described above and the gDNA of these strains was extracted using the Plant DNeasy mini kit (Qiagen, Marseille, France) as per the manufacturer’s instructions, then kept at −20 °C until further use.

Conventional and real-time PCR reaction conditions

The combination of the forward and reverse primers, and the probe developed in this project will be referred to cCBS (for conventional PCR) and qCBS (for real-time PCR): CBS stands for citrus black spot. Conditions for both cCBS and qCBS were assessed to ascertain their specificity towards the target species P. citricarpa. In particular, the assays were optimized not to cross-react with DNA of P. paracitricarpa or P. citriasiana, which are genetically closely related to P. citricarpa.

cCBS reactions contained 1× PCR reaction buffer, 1.5 mM of MgCl2, 0.25 mM of each dNTP, 0.6 µM of each forward (cCBS-F) and reverse (cCBS-R) primer, 0.05 U of HGS Diamond Taq DNA polymerase (Eurogentec, Seraing, Belgium), 0.6 µg/µL Bovine Serum Albumin (BSA) and 2 µL of DNA template. Molecular grade water was added to the reaction up to 20 µL. cCBS PCR runs were performed using an initial denaturation step at 95 °C for 10 min, followed by 40 cycles of denaturation at 94 °C for 30 s, annealing at 64 °C for 30 s and elongation at 72 °C for 45 s, with a final elongation step at 72 °C for 10 min. cCBS PCR runs were performed in a BioRad T100 thermocycler (Bio-Rad, Hercules, CA, USA). cCBS PCR products were visualized in a 1.5% agarose gel in 0.5× tris-borate-EDTA buffer stained with Ethidium bromide.

qCBS reactions contained 1× PCR reaction buffer, 5 mM of MgCl2, 0.2 mM of each dNTP, 0.1 µM of forward primer qCBS-F, 0.2 µM of reverse primer qCBS-R, 0.1 µM of qCBS-P probe, 0.025 U µL of the qPCR Core Kit DNA polymerase, 0.6 µg/µL Bovine Serum Albumin (BSA), and 2 µL of DNA template. Molecular grade water was added up to 20 µL for the final volume. The qCBS cycling conditions were 1 cycle of initial denaturation at 95 °C for 10 min, and 45 cycles of denaturation at 95 °C for 15 s followed by annealing at 68 °C for 1 min. qCBS reactions were performed using the Eurogentec Core Kit No ROX in a Rotor-Gene 6500 thermocycler (Qiagen, Marseille, France). Each reaction’s Ct values and the standard deviations obtained from all the samples’ replicates were recorded by the Rotor-Gene Q series software (v 2.3.5; Qiagen, Marseille, France).

Construction of a plasmid positive control for conventional PCR and real-time PCR

The gDNA of P. citricarpa strain LSVM 1501 was used to produce the plasmid positive control for the primer pairs qCBS-F (cCBS-F) and qCBS-R (cCBS-R), using the Clone JET PCR Cloning kit (Thermo Fisher Scientific, Illkirch-Graffenstaden, France). Amplicons generated with the primers were inserted in the pJET1.2/blunt vector in order to transform competent DH10B cells of Escherichia coli. These E. coli cells were then transferred to Petri dishes containing Luria-Bertani (LB) broth amended with 50 mg L−1 of ampicillin, and were incubated overnight at 37 °C. The plasmid DNA (pDNA) was extracted from the bacterial cells using the Nucleospin Plasmid kit (Macherey-Nagel, Düren, Germany). The molecular mass of the plasmid and the number of plasmid copies (pc) produced were calculated, and a dilution series of the plasmid was further tested by cCBS and qCBS. Plasmid DNA (pDNA) was diluted in 1× tris-EDTA (TE) buffer and kept at −20 °C until use.

Performance assessment of the optimized conventional and real-time PCR assays

The analytical specificity of the cPCR and qPCR assays was assessed with a DNA panel that included 30 non-target Phyllosticta spp. and 21 other fungal species isolated from citrus fruits (Table 1). The inclusivity of both cCBS and qCBS assays was assessed with a panel of 26 P. citricarpa isolates from different hosts and origins (Table 1). All gDNA samples were diluted at 1 ng µL−1. Runs included two replicates per sample. Negative controls were included in each run and consisted of DNA extracted from healthy orange and lemon peel and normalized at 1 ng µL−1.

The analytical sensitivity of the cCBS and qCBS assays was assessed by testing a ten-fold dilution series of the pDNA diluted in 1× tris-EDTA (TE) buffer and a ten-fold dilution series of the pDNA diluted in a background of 1 ng µL−1 of healthy orange and lemon peel (flavedo) DNA, mixed at a ratio of 1:1. Each dilution series ranged from 3.16 106 to 3.16 pc. The limit of detection (LOD) was estimated as the lowest concentration of pDNA in 1× TE buffer that yielded 100% positive results on all replicates included in the cCBS and qCBS reactions in our conditions. In the case of qCBS, a standard curve was obtained for each type of matrix tested (pDNA in 1× TE buffer, pDNA with orange peel DNA and lemon peel DNA).

The qCBS assay was further assessed by evaluating additional criteria such as repeatability, reproducibility, robustness and transferability, in order to check its behavior in conditions close to routine analysis with target and non-target DNA. All these four criteria were evaluated with a panel consisting of pDNA at 10×, 100× and 1,000× (only for repeatability) the LOD, as well as 0.1 ng µL−1 of gDNA of P. citricarpa strain LSVM1501 and diluted in 1 ng µL−1 solution of healthy orange or lemon peel DNA (ratio 1:1), gDNA of P. paracitricarpa strains ZJUCC200937 and LSVM 1238, and P. citriasiana strain LSVM 1146 at 1 ng µL−1 each.

For repeatability, ten replicates of each template were tested by a single operator using the same real-time PCR equipment (Rotorgene Q; Qiagen, Marseille, France). For reproducibility, three replicates of each template were tested on the same qPCR equipment by two different operators over 3 days. The robustness of the qPCR assay was evaluated by modifying two qPCR parameters. First, the qPCR assay was performed using a ±10% variation in the final qPCR reaction volume (i.e., 18 and 22 µL). Second, the qPCR assay was performed by modifying the hybridization temperature by ± 2 °C (i.e., 66 °C and 70 °C). These qPCR assays were performed with 10 replicates of each template.

The transferability of the method was assessed by comparing the performance of two different thermocyclers (Rotorgene Q and Roche Lightcycler 480) and four different qPCR kits or master mixes (No ROX qPCR Core Kit, Eurogentec; QuantaBio PerfeCTa qPCR ToughMix; No ROX qPCR Master Mix, Eurogentec; and the Eurogentec Takyon Core kit) in the same laboratory. An additional experiment was added by involving another laboratory (IAM-GIHF, Madrid, Spain), which used its own master mix (TaKaRa Premix Ex Taq™ (Probe qPCR); TaKaRa, Kusatsu, Japan) and equipment (Rotorgene Q; Qiagen, Marseille, France). The qPCRs were carried out using ten replicates of each template.

Analysis of naturally infected citrus materials

The cCBS PCR and qCBS qPCR assays were assessed on a set of citrus fruit samples with symptoms that looked like citrus black spot. It included 107 DNA samples comprising DNAs previously extracted from naturally infected citrus fruits of various provenances and species intercepted in French harbors between 2018–2020, as well as 12 DNAs obtained from fruit lesions/symptoms that resembled CBS disease but tested negative for P. citricarpa with van Gent-Pelzer et al.’s (2007) assay (Table 3). These 12 DNA samples were used as negative controls. In addition, DNA samples extracted from CBS lesions on infected fresh fruits obtained for this study from Argentina and two sites in Tunisia (Table 3) were also tested. DNA was extracted from CBS lesions using the DNeasy Plant Mini kit (Qiagen, Marseille, France) following the manufacturer’s protocol. The initial grinding step was performed using Lysing Matrix A tubes and a FastPrep24 grinding machine (MP Biomedicals, Santa Ana, CA, USA), with a two-run program at 6.5 U for 1 min.

Table 3 Testing of citrus fruits showing CBS-like symptoms by the cCBS PCR, qCBS qPCR, and van Gent-Pelzer et al. (2007) qPCR.

Sample	Year	Host	Organ	Origin	cCBS PCR	qCBS qPCR	VGP qPCR	18S uni qPCR	
18-429/2a	2018	Citrus reticulata × C. sinensis	Fruit	Argentina	+	28.74 ± 0.34	18.78 ± 0.17	11.67 ± 0.00	
18-429/2b	2018	Citrus reticulata × C. sinensis	Fruit	Argentina	+	29.01 ± 0.04	20.07 ± 0.00	11.97 ± 0.10	
18-429/2c	2018	Citrus reticulata × C. sinensis	Fruit	Argentina	+	28.87 ± 0.01	20.38 ± 0.04	12.23 ± 0.05	
18-429/2d	2018	Citrus reticulata × C. sinensis	Fruit	Argentina	+	27.63 ± 0.29	17.37 ± 0.09	10.74 ± 0.06	
18-429/2e	2018	Citrus reticulata × C. sinensis	Fruit	Argentina	+	27.71 ± 0.53	18.06 ± 0.08	10.94 ± 0.04	
18-429/2f	2018	Citrus reticulata × C. sinensis	Fruit	Argentina	n.t.	30.41 ± 2.34	19.94 ± 0.01	10.76 ± 0.05	
18-429/2g	2018	Citrus reticulata × C. sinensis	Fruit	Argentina	+	32.69 ± 0.29	23.54 ± 0.07	12.79 ± 0.04	
18-429/4a	2018	Citrus × limonia	Leaves	Argentina	n.t.	24.16 ± 0.29	16.94 ± 0.22	15.37 ± 0.13	
18-429/4b	2018	Citrus × limonia	Leaves	Argentina	+	28.61 ± 1.26	14.02 ± 0.91	12.00 ± 0.02	
18-429/4c	2018	Citrus × limonia	Leaves	Argentina	+	24.05 ± 0.25	16.44 ± 0.33	13.80 ± 0.11	
18-429/4d	2018	Citrus × limonia	Leaves	Argentina	+	23.55 ± 0.02	15.20 ± 0.08	11.39 ± 0.30	
18-429/4e	2018	Citrus × limonia	Leaves	Argentina	+	24.59 ± 0.31	16.52 ± 0.21	12.68 ± 0.11	
18-429/4f	2018	Citrus × limonia	Leaves	Argentina	+	33.03 ± 0.47	24.99 ± 0.02	15.94 ± 0.11	
18-429/4g	2018	Citrus × limonia	Leaves	Argentina	+	27.30 ± 0.18	19.39 ± 0.03	14.84 ± 0.03	
18-429/5a	2018	Citrus limon	Fruit	Argentina	+	31.78 ± 0.04	23.16 ± 0.05	12.47 ± 0.04	
18-429/5b	2018	Citrus limon	Fruit	Argentina	+	31.28 ± 0.01	21.92 ± 0.10	12.35 ± 0.02	
18-429/5c	2018	Citrus limon	Fruit	Argentina	+	30.03 ± 0.26	21.57 ± 0.01	11.63 ± 0.00	
18-429/5d	2018	Citrus limon	Fruit	Argentina	+	29.58 ± 0.08	21.03 ± 0.19	12.51 ± 0.10	
18-429/5e	2018	Citrus limon	Fruit	Argentina	+	29.30 ± 0.12	20.69 ± 0.03	12.45 ± 0.02	
18-429/5f	2018	Citrus limon	Fruit	Argentina	+	32.10 ± 0.16	22.86 ± 0.02	14.42 ± 0.04	
18-429/5g	2018	Citrus limon	Fruit	Argentina	+	30.03 ± 0.03	21.56 ± 0.03	13.52 ± 0.08	
18-429/6a	2018	Citrus sinensis	Fruit	Argentina	+	28.56 ± 0.09	19.98 ± 0.01	9.96 ± 0.06	
18-429/6b	2018	Citrus sinensis	Fruit	Argentina	+	28.40 ± 0.30	19.67 ± 0.05	9.00 ± 0.05	
18-429/6c	2018	Citrus sinensis	Fruit	Argentina	+	30.37 ± 0.39	21.90 ± 0.02	9.69 ± 0.03	
18-429/6d	2018	Citrus sinensis	Fruit	Argentina	+	30.76 ± 0.15	21.99 ± 0.14	8.99 ± 0.11	
18-429/6e	2018	Citrus sinensis	Fruit	Argentina	+	30.42 ± 0.08	21.42 ± 0.13	9.47 ± 0.02	
18-429/6f	2018	Citrus sinensis	Fruit	Argentina	+	31.68 ±0.09	22.80 ± 0.06	8.82 ± 0.14	
18-429/6g	2018	Citrus sinensis	Fruit	Argentina	+	33.13 ± 0.36	24.12 ± 0.05	9.77 ± 0.08	
18-349	2018	Citrus limon	Fruit	Argentina	+	31.57 ± 0.43	23.78 ± 0.01	12.39 ± 0.11	
18-374	2018	Citrus limon	Fruit	Argentina	+	35.13 ± 1.00	27.45 ± 0.08	14.35 ± 0.01	
18-460/1	2018	Citrus sinensis	Fruit	Brazil	+	28.77 ± 0.35	20.82 ± 0.15	11.61 ± 0.01	
18-551	2018	Citrus sinensis	Fruit	Benin	+	33.03 ± 0.17	22.48 ± 0.11	11.18 ± 0.54	
18-552	2018	Citrus sp.	Fruit	Benin	+	31.06 ± 0.18	20.21 ± 0.01	9.25 ± 0.32	
19-065	2019	Citrus limon	Fruit	Tunisia	n.t.	32.13 ± 0.45	21.23 ± 0.01	8.72 ± 0.06	
19-116	2019	Citrus sinensis	Fruit	Tunisia	+	28.82 ± 0.09	17.79 ± 0.14	8.31 ± 0.27	
19-117	2019	Citrus sinensis	Fruit	Tunisia	–	>45	35.78*,a	8.77 ± 0.05	
19-118	2019	Citrus sinensis	Fruit	Tunisia	+	28.90 ± 0.14	18.68 ± 0.01	9.77 ± 0.00	
19-119	2019	Citrus sinensis	Fruit	Tunisia	+	34.56 ± 0.18	23.25 ± 0.07	8.57 ± 0.02	
19-133	2019	Citrus limon	Fruit	Tunisia	+	24.32 ± 0.47	13.43 ± 0.11	10.79 ± 0.05	
19-146/a	2019	Citrus sinensis	Fruit	Tunisia	+	29.53 ± 0.09	17.97 ± 0.02	7.92 ± 0.02	
19-146/b	2019	Citrus sinensis	Fruit	Tunisia	+	31.74 ± 0.24	20.65 ± 0.01	7.86 ± 0.03	
19-159/2	2019	Citrus limon	Fruit	Tunisia	–	>45	>45	10.72 ± 0.04	
19-5150/1	2019	Citrus sinensis	Fruit	South Africa	+	31.28 ± 0.17	19.66 ± 0.12	8.31 ± 0.36	
19-5150/2	2019	Citrus sinensis	Fruit	South Africa	+	30.59 ± 0.09	19.51 ± 0.18	8.66 ± 0.04	
19-5150/3	2019	Citrus sinensis	Fruit	South Africa	+	32.37 ± 0.07	20.24 ± 0.01	7.65 ± 0.01	
19-5150/4	2019	Citrus sinensis	Fruit	South Africa	+	30.74 ± 0.10	19.64 ± 0.07	9.71 ± 0.07	
19-5150/5	2019	Citrus sinensis	Fruit	South Africa	+	33.11 ± 0.25	21.83 ± 0.05	8.85 ± 0.08	
19-5241/1	2019	Citrus sinensis	Fruit	South Africa	+	36.27 ± 0.41	24.45 ± 0.05	11.32 ± 0.08	
19-5241/2	2019	Citrus sinensis	Fruit	South Africa	+	31.37 ± 0.14	20.25 ± 0.10	8.83 ± 0.01	
20-2990	2020	Citrus sinensis	Fruit	Brazil	+	28.68 ± 0.05	18.46 ± 0.12	9.60 ± 0.00	
20-2999/1	2020	Citrus limon	Fruit	Argentina	+	35.02 ± 0.12	22.54 ± 0.06	9.96 ± 0.08	
20-3282/1	2020	Citrus limon	Fruit	Argentina	–	>45	35.04 ± 0.03a	8.49 ± 0.02	
20-2382/2	2020	Citrus limon	Fruit	Argentina	+	32.74 ± 0.16	22.55 ± 0.27	8.52 ± 0.02	
20-3493/1	2020	Citrus limon	Fruit	Argentina	+	30.29 ± 0.03	20.05 ± 0.12	9.43 ± 0.01	
20-3493/2	2020	Citrus limon	Fruit	Argentina	+	35.81 ± 0.04	24.43 ± 0.18	9.78 ± 0.05	
20-3493/3	2020	Citrus limon	Fruit	Argentina	+	33.82 ± 0.20	22.75 ± 0.11	9.40 ± 0.01	
20-3493/4	2020	Citrus limon	Fruit	Argentina	+	33.75 ± 0.16	22.79 ± 0.01	9.95 ± 0.02	
20-3493/5	2020	Citrus limon	Fruit	Argentina	+	33.42 ± 0.08	22.50 ± 0.03	9.87 ± 0.03	
20-3504/1	2020	Citrus limon	Fruit	Argentina	+	31.69 ± 0.28	21.34 ± 0.02	8.70 ± 0.03	
20-3504/2	2020	Citrus limon	Fruit	Argentina	+	32.83 ± 0.01	22.03 ± 0.00	9.84 ± 0.03	
20-3504/4	2020	Citrus limon	Fruit	Argentina	+	36.22 ± 0.34	24.63 ± 0.04	8.64 ± 0.01	
20-3504/5	2020	Citrus limon	Fruit	Argentina	+	31.95 ± 0.27	21.58 ± 0.18	9.32 ± 0.06	
20-0479	2020	Citrus reticulata	Fruit	South Africa	+	32.24 ± 0.04	22.41 ± 0.08	12.36 ± 0.04	
TN1-F01	2021	Citrus limon	Fruit	Tunisia	+	32.62 ± 0.22	22.56 ± 0.13	10.64 ± 0.03	
TN1-F02	2021	Citrus limon	Fruit	Tunisia	n.t.	36.10 ± 0.92	25.50 ± 0.05	9.89 ± 0.01	
TN1-F03	2021	Citrus limon	Fruit	Tunisia	+	34.12 ± 0.12	23.04 ± 0.03	8.89 ± 0.00	
TN1-F04	2021	Citrus limon	Fruit	Tunisia	+	35.36 ± 0.28	24.35 ± 0.03	10.26 ± 0.30	
TN1-F05	2021	Citrus limon	Fruit	Tunisia	+	33.46 ± 0.27	22.57 ± 0.01	9.68 ± 0.01	
TN1-F06	2021	Citrus limon	Fruit	Tunisia	+	34.17 ± 0.41	23.80 ± 0.07	10.55 ± 0.02	
TN1-F07	2021	Citrus limon	Fruit	Tunisia	+	35.35 ± 0.44	24.63 ± 0.03	8.73 ± 0.06	
TN1-F08	2021	Citrus limon	Fruit	Tunisia	+	36.39 ± 0.53	25.14 ± 0.00	10.60 ± 0.01	
TN1-F09	2021	Citrus limon	Fruit	Tunisia	+	32.31 ± 0.39	22.14 ± 0.11	11.79 ± 0.02	
TN1-F10	2021	Citrus limon	Fruit	Tunisia	+	32.43 ± 0.22	22.41 ± 0.01	10.67 ± 0.06	
TN1-F11	2021	Citrus limon	Fruit	Tunisia	+	32.16 ± 0.52	21.95 ± 0.00	10.61 ± 0.06	
TN1-F12	2021	Citrus limon	Fruit	Tunisia	+	33.23 ± 0.17	23.06 ± 0.01	10.50 ± 0.08	
TN2-F01	2021	Citrus limon	Fruit	Tunisia	+	32.48 ± 0.15	23.31 ± 0.00	9.68 ± 0.13	
TN2-F05	2021	Citrus limon	Fruit	Tunisia	+	34.95 ± 0.10	24.83 ± 0.01	10.62 ± 0.10	
AR-F01	2021	Citrus sinensis	Fruit	Argentina	+	33.43 ± 0.14	23.52 ± 0.01	11.55 ± 0.17	
AR-F02	2021	Citrus sinensis	Fruit	Argentina	+	34.54 ± 0.56	24.07 ± 0.03	11.86 ± 0.20	
AR-F03	2021	Citrus sinensis	Fruit	Argentina	+	31.98 ± 0.20	22.83 ± 0.05	12.19 ± 0.01	
AR-F04	2021	Citrus sinensis	Fruit	Argentina	+	33.34 ± 0.51	22.92 ± 0.04	11.28 ± 0.00	
AR-F05	2021	Citrus sinensis	Fruit	Argentina	+	32.24 ± 0.42	22.64 ± 0.05	11.20 ± 0.02	
AR-F06	2021	Citrus sinensis	Fruit	Argentina	+	32.08 ± 0.10	22.42 ± 0.06	12.27 ± 0.09	
20-01348	2020	Citrus sinensis	Fruit	Tunisia	–	>45	>45	8.87 ± 0.11	
20-01427	2020	Citrus reticulata	Fruit	Israel	–	>45	>45	10.46 ± 0.04	
20-04070	2020	Citrus sinensis	Fruit	Egypt	–	>45	>45	10.87 ± 0.03	
20-02794	2020	Citrus limon	Fruit	South Africa	–	>45	>45	9.32 ± 0.19	
20-02836	2020	Citrus limon	Fruit	Argentina	–	>45	32.21 ± 0.90a	9.95 ± 0.02	
20-05365	2020	Citrus sp.	Fruit	Japan	–	>45	>45	11.59 ± 0.13	
21-00466	2021	Citrus sp.	Fruit	Bangladesh	–	>45	>45	8.51 ± 0.06	
21-01914	2021	Citrus latifolia	Fruit	Brazil	–	>45	>45	8.74 ± 0.02	
21-03392	2021	Citrus reticula	Fruit	Egypt	–	>45	>45	12.44 ± 0.03	
333$	2015	Citrus sinensis	Fruit	Brazil	+	25.02 ± 0.03	19.77 ± 0.02	n.t	
483$	2018	Citrus limon	Fruit	Argentina	+	25.27 ± 0.12	20.13 ± 0.09	n.t	
507$	2018	Citrus sinensis	Fruit	Argentina	+	25.15 ± 0.34	19.74 ± 0.09	n.t	
546$	2018	Citrus sinensis	Fruit	Brazil	+	24.78 ± 0.07	19.84 ± 0.14	n.t	
599$	2019	Citrus limon	Fruit	Argentina	+	22.97 ± 0.41	18.46 ± 0.01	n.t	
626$	2019	Citrus reticulata	Fruit	Uruguay	+	24.14 ± 0.08	18.14 ± 0.10	n.t	
647$	2019	Citrus sinensis	Fruit	Uruguay	+	25.11 ± 0.05	20.14 ± 0.07	n.t	
686$	2020	Citrus limon	Fruit	Argentina	+	29.08 ± 0.45	24.18 ± 0.06	n.t	
703$	2020	Citrus limon	Fruit	Uruguay	+	27.86 ± 0.05	22.65 ± 0.06	n.t	
816$	2020	Citrus sinensis	Fruit	Uruguay	+	23.9 ± 0.23	18.63 ± 0.02	n.t	
890$	2021	Citrus sinensis	Fruit	Zimbabwe	+	25.24 ± 0.09	20.39 ± 0.07	n.t	
897$	2021	Citrus limon	Fruit	South Africa	+	24.33 ± 0.10	18.68 ± 0.05	n.t	
907$	2021	Citrus sinensis	Fruit	South Africa	+	26.95 ± 0.02	22.29 ± 0.03	n.t	
1029$	2022	Citrus limon	Fruit	Argentina	+	27.9 ± 0.47	23.06 ± 0.01	n.t	
1069$	2022	Citrus sinensis	Fruit	Argentina	+	27.45 ± 0.06	22.16 ± 0.07	n.t	
Notes:

The 18S uni test was conducted to verify the amplifiability of the DNA extracts. The mean and standard deviation of Ct values were generated with two replicates. A value flagged with * means that only one out of the two replicates was positive. N.t stands for not tested.

$ DNA extracted from fruit lesions by the IAM-GIHF laboratory.

a Sample classified as negative according to our internal rules for the VGP assay.

All DNA extracts were also analyzed using the qPCR method of van Gent-Pelzer et al. (2007), used as the gold standard reference method, and hereafter referred to as VGP assay. In addition, the 18S Uni test developed by Ioos et al. (2009) that targets the 18S rDNA of plants and fungi by qPCR, was used to assess the quality of the amplified DNAs. Two replicates of each sample were included in the assays. The relative sensitivity, specificity and accuracy of the newly developed cCBS and qCBS were calculated according to ISO 16140 (International Standardization Organization, 2016) and Ioos & Iancu (2008).

Results

Phylogeny of Phyllosticta spp from Citrus deduced from single-copy genes

The protein sequences searched for in the annotated genomes of P. citricarpa allowed 51 genes to be selected with a single hit. Alignments of each of these genes separately revealed that 14 (27.5%) were polymorphic and could be used to discriminate between strains of P. citricarpa and P. paracitricarpa. In contrast, 28 genes (54.9%) were completely monomorphic in the two species. Finally, nine genes (17.6%) exhibited some intra-species polymorphism, but were not fixed in either all the P. citricarpa or all the P. paracitricarpa strains.

Concatenated sequences of these 51 genes resulted in a 84,155-bp alignment. In all, 35 polymorphic sites differentiated all the strains of P. citricarpa from all those of P. paracitricarpa. The average percentage relatedness between the 51 concatenated genes from P. citricarpa and P. paracitricarpa was 99.26% (±0.89 SD). The phylogenetic tree pattern was in general consistent with previous identification of Phyllosticta strains using LSU and tef1: strains were assigned to separate clades corresponding to P. capitalensis, P. citrichinaensis, P. citribraziliensis, P. citriasiana, P. paracitricarpa, and P. citricarpa (Fig. 1). In line with its microsatellite multilocus genotype pattern (Information S3), LSVM1238 grouped with the clade including all the P. paracitricarpa reference strains. Therefore, strain LSVM1238 was assigned tentatively to P. paracitricarpa and referred to as such in this work (Table 1).

Figure 1 Maximum likelihood and Bayesian phylogenetic cladogram.

The cladogram was constructed based on 51 single-copy contatenated genes obtained from the FunyBase database (Marthey et al., 2008). The bayesian posterior probabilities (>0.9) and bootstrap (>70) support values are indicated alongside the branches of the tree.

Selection of a region highly specific to P. citricarpa and design of specific oligonucleotides

A region showing fixed polymorphisms between the sister species P. citricarpa and P. paracitricarpa was successfully identified in scaffold_23 of the P. citricarpa CBS 102373 genome assembly, between base positions 516,600 and 517,100. In particular, when compared with genomes of P. citricarpa and P. citriasiana a 7-pb insertion (located between base positions 516,829 and 516,830) and a 7-pb deletion (located between base positions 516,848 and 516,854) was present in all P. paracitricarpa genomes included in the selection (Fig. 2). Tentative primers (for cPCR and qPCR) and a probe (for qPCR) able to distinguish P. citricarpa from P. paracitricarpa were designed targeting both the insertion and the deletion sequences and a few SNPs located upstream. These primers and probes were also designed to be able to distinguish P. citricarpa from P. citriasiana. One SNP between these two species is located in the forward primer (cCBS-F or qCBS-F), one in the probe (qCBS-P), and one in the reverse primer (cCBS-F or qCBS-F) (Fig. 2).

Figure 2 Alignment of representative sequences of Phyllosticta citricarpa, P. paracitricarpa and P. citriasiana of the gene located in the scaffold 23 of the P. citricarpa CBS102373 strain genome.

The sequences of the reference genome are highligthed in yellow. The primers for conventional (cCBS-F/cCBS-R) and real-time PCR (qCBS-F/qCBS-R) are flagged in green and yellow. The probe of the real-time PCR assay (qCBS-P) is flagged in red. SNPs and indels located in the primers or probe’s target regions are highlighted in blue.

Performance values of the assays

Using the conditions and parameters listed in the materials and methods section, both cPCR and qPCR assays successfully amplified DNA from all 26 P. citricarpa strains, regardless of their geographic provenance, and did not cross-react with any of the non-target species, including the sister species P. paracitricarpa and closely related species P. citriasiana. The accuracy of negative results yielded with cCBS and qCBS assays for all the DNAs was checked by successfully amplifying the ITS using the fungal universal ITS5/ITS4 PCR test (Table 2).

Regarding the analytical sensitivity, the limit of detection obtained for qPCR assays was 31.6 pc µL−1 using the pDNA diluted in 1× TE buffer as well as for the pDNA diluted in a background of 1 ng µL−1 of the DNA from healthy orange and lemon peel. The LOD for cPCR was higher, with 316 pc µL−1 in all conditions. For the cPCR assay, 100% of the samples tested at that concentration yielded a positive result with an amplicon size of 107 bp. For the qPCR assay, the R2 values of the pDNA in 1× TE buffer, and the pDNA diluted in orange and lemon peel DNA were 0.995, 0.996, and 0.995, respectively (Fig. 3). In all further qCBS runs, a LOD positive control was included in duplicate, and its mean Ct value was used as the cut-off value to declare a DNA extract positive or negative regarding P. citricarpa.

Figure 3 Standard curves assessed with a 10-fold serial dilution of the P. citricarpa plasmid DNA positive control.

The plasmid DNA standards were diluted in 1× Tris-EDTA (TE) buffer (circle), in a background of lemon DNA (triangle), or in a background of orange DNA (square). The mean Ct values are calculated with three replicates

The remaining performance values, repeatability, reproducibility, robustness and transferability are summarized in the supplemental sections. Both conventional and real-time PCR assays yielded 100% repeatable and reproducible results (Information S4). The coefficient of variation for repeatability ranged between 0.53% and 1.37%. The coefficient of variation for reproducibility ranged between 3.49% and 4.41%. Non-target DNAs were never amplified, as expected.

The robustness of the qPCR method was assessed by modifying the reaction volume of ±10% and the hybridization temperature of ±2 °C. Modifying the reaction volume (18 and 22 µL instead of 20 µL) resulted in a few late Ct values with the DNA of P. citriasiana (isolate LSVM1146), indicative of cross-reactions. All these amplifications had Ct values over 40 (Information S4). Changes in the hybridization temperature (66 °C and 70 °C instead of 68 °C) affected either sensitivity or specificity. In less stringent conditions at 66 °C, late Ct values (35.28 ± 0.52) were obtained with P. citriasiana DNA. However, when the hybridization/polymerization temperature was increased to 70 °C, the qPCR failed to amplify the target close to the limit of detection (10× LOD and 100× LOD), and generated inconsistently delayed Ct values with the target DNA diluted in orange or lemon peel (Information S5).

The transferability of the qPCR method was assessed by comparing five different qPCR master mixes and two qPCR platforms. The No ROX qPCR Core Kit (Eurogentec, Seraing, Belgium) and Takyon No ROX Probe Core Kit (Eurogentec, Seraing, Belgium) yielded satisfactory results in terms of specificity. However, changing the qPCR master mix sometimes compromised the assay’s sensitivity and specificity (Information S6). First, the Eurogentec No ROX qPCR Master Mix (Eurogentec, Seraing, Belgium) and the TaKaRa Premix Ex Taq Probe qPCR (runs carried out by IAM) did not amplify any of the positive samples included in the run. Second, the Takyon No ROX Probe Core Kit (Eurogentec, Seraing, Belgium) yielded cross-reactions with the DNA of P. citriasiana (isolate LSVM1146). Third, the QuantaBio PerfeCTa qPCR ToughMix yielded late Ct amplifications of the target DNA (>35 Ct). Regarding the qPCR platform, runs performed with the Roche Light Cycler 480 using optimized conditions were generally similar to those using the Rotorgene thermocycler.

Tests on fruits with CBS symptoms

Fruits of different origins and varieties were analyzed with both the cCBS and the qCBS assays. These analyses were compared with those of the VGP reference method (VGP qPCR). All DNA extracts yielding negative results with all three P. citricarpa assays were also tested with 18S uni qPCR (Ioos et al., 2009) in order to verify the quality of the DNA extract (Table 3).

Out of the 107 DNAs included, both qPCR assays yielded 100% identical results with 95 positive samples and 12 negative results (Table 3). The end-point cCBS PCR provided identical results except for three samples which could not be tested. On two occasions, the VGP qPCR assay yielded a late mean Ct value (>35), whereas cCBS and qCBS assay results were negative. However, according to our internal rules these late Ct values were not considered as positive, since they were later than the mean Ct generated with the VGP assay’s limit of detection control. Considering the VGP qPCR assay as the reference method (EPPO, 2020), the relative sensitivity, specificity and accuracy of the cCBS cPCR and qCBS qPCR assays were all 100%.

The mean Ct value obtained for the whole set of positive samples was significantly higher with the qCBS qPCR (30.52 ± 3.4), targeting a single-copy region, than with the VGP assay (21.51 ± 3.1) that targets a multi copy operon.

Discussion

In this work, genomes from different Phyllosticta species were used both to further examine their phylogenetic relationships, and to develop a more specific assay targeting the regulated species P. citricarpa. This perfectly illustrates the power of comparative genomics and the support it provides in managing priority fungal plant pathogens. Indeed, a precise identification of the pathogen is a prerequisite first to draft regulations, which refer to specific taxon names, and second to develop and use an accurate detection test for early and specific detection in order to prevent the pathogen’s introduction into a country or to attempt to eradicate it as early as possible once detected for the first time. This strategy may also apply to other high priority pathogens.

The first achievement of this work is the confirmation that the strains of P. citricarpa and P. paracitricarpa included in this work group in two distinct phylogenetic clades. A multilocus alignment of 51 concatenated genes is in line with the results of Guarnaccia et al. (2017) and Wang et al. (2023). The phylogenies of Guarnaccia et al. (2017) and Wang et al. (2023) were inferred using six genes from the core genome. Guarnaccia et al. (2019) have also used whole genomes to assess the genetic relatedness between the species. All these studies show a separation of the two phylogenetic clades. Additionally, the same clustering pattern was observed on a subset of ten strains by microsatellite genotyping. Our work demonstrates that genetic differentiation between the two clades is very low (>99%), with more than 50% of the genes included that were monomorphic. However, a certain level of genetic structure occurs within each clade, as illustrated by the support of the branches in the phylogenetic tree. Although beyond the scope of this work, the inclusion of a larger number of strains from different origins, combined with analysis of genealogical concordance (Gladieux et al., 2018), would be necessary to accurately assess the taxonomy of P. citricarpa and P. paracitricarpa.

One important result of our work concerns the (un)reliability of using the tef1 gene to identify some isolates of P. citricarpa and P. paracitricarpa. Indeed, our phylogeny, together with microsatellite genotyping and the negative testing of its DNA with qCBS and cCBS PCR argue in favor of the assignation of the Bangladeshi strain LSVM1238 to the P. paracitricarpa clade. P. paracitricarpa has intra-species polymorphisms in tef1 (Information S7), whereas “hallmarks” of P. paracitricarpa were observed in P. citricarpa, although having a low allele frequency. This is in line with the pattern observed for nine genes (17.6%) used in our phylogeny study. Therefore, utilization of tef1 in the context of quarantine testing can present a risk of false positives.

The second output of our comparative genomics was the development and validation of new conventional and real-time PCR assays to accurately detect the CBS-causing agent Phyllosticta citricarpa. This species is the only Phyllosticta species which is internationally regulated, being a quarantine fungus for numerous regional plant protection organizations in the world. Other species are also able to cause spots on citrus fruits, such as P. citriasiana, P. citrimaxima, P. citrichinaensis (Guarnaccia et al., 2019), and the recently described sister species, P. paracitricarpa. This latter species was recently found to have a wider host and geographic distribution on Citrus spp. in China (Wang et al., 2023). Therefore, the protocols for detecting the only quarantine species, P. citricarpa, must be sufficiently specific to avoid any cross-reaction with these other pathogenic species, as well as with endophytic species occurring on fruits or leaves. To our knowledge, all the current P. citricarpa detection protocols based on molecular assays targeting the ITS rDNA region must produce false positive results with P. paracitricarpa due to their genetic closeness. This is in particular the case with the PCR and qPCR assays of van Gent-Pelzer et al. (2007), Schirmacher et al. (2019), and Peres et al. (2007), which are recommended in international standards (EPPO, 2020; IPPC, 2014).

In addition, while proceeding through the steps of comparative genomics conducted in this work, it was observed that the sequence of the housekeeping gene tef1 varied across the five P. paracitricarpa isolates. This contradicts reports saying that some of the SNPs in tef1 could be used to unequivocally identify P. citricarpa by sequencing (EPPO, 2020) or qPCR (Zajc et al., 2022), but our own observations suggest that these methods should be used with caution, since tef1 from some P. paracitricarpa strains may display the “hallmarks” of P. citricarpa (Information S7).

The comparative genomics approach reported here screened regions that were highly polymorphic from one sister species to another. The region located on P. citricarpa genome scaffold_23 was also present in P. paracitricarpa, but conserved at the intraspecific level while showing a high degree of interspecific polymorphism. It was therefore fit for the design of specific oligonucleotides.

The specificity of an assay targeting a quarantine organism should be thoroughly verified and validated. First, it should be stressed that a comprehensive and representative panel of strains or genomes is required for the validation step, in order to cover as much natural genetic diversity as possible. Even with these precautions, the test’s specificity must be constantly checked, as new strains with a slightly different genetic background may be discovered over time. The observation of unexpected variation in tef1 within P. citricarpa is a good example, and variation in the region located on P. citricarpa genome scaffold_23 may never be ruled out as a possibility. In this respect, it is of utmost importance to report the occurrence of new strains that cause unexpected results (Gupta et al., 2018), especially when related to quarantine pests involving severe regulatory measures. When feasible, it is also important to include several other species from the same genus during the optimization/validation steps, and not only the closest species. Hence, in our work to design oligonucleotides that could discriminate the closely related P. citricarpa from P. paracitricarpa, more stringent reaction conditions had to be applied to prevent cross-amplification of the DNA of the more distant species, P. citriasiana.

The validation results provided herein support the need to carry out transferability and robustness studies, especially when stringency conditions are very strict because there is a need to discriminate between closely related species (Ioos et al., 2019). In the course of this work, an assessment of other commercial qPCR master mixes showed that some of them dramatically reduced the DNA amplification yield or hampered hybridization of the primers or probe, thus leading to an unacceptable loss of sensitivity in the qCBS qPCR assay. It is therefore essential before using the tests routinely and for regulatory analyses, to ensure that under new laboratory conditions, and using commercial reagents that may differ from those used in the original work, the reaction remains sufficiently sensitive and specific. Overall, in our conditions and with our reagents, an increase in the annealing temperature to 70 °C was the only deviation that generated false negative results, except with P. citricarpa gDNA diluted in orange peel DNA. This high temperature significantly affected the sensitivity, probably due to loss of DNA polymerase efficiency or poor probe hybridization. This is also probably the reason why no Ct value could be generated with the panel of samples analyzed by IAM, using a qPCR master mix (TaKaRa Premix Ex Taq ™ Probe qPCR; TaKaRa, Kusatsu, Japan) containing a DNA polymerase which the supplier recommends should be used at a polymerization temperature range of 56–64 °C. Likewise, in our study, changing the qPCR master mix brand or type also affected the specificity of the VCG qPCR assay, since false-positive results were frequently obtained with the DNA of P. citriasiana.

In addition, non-target DNA from P. citriasiana strain LSVM 1146 was occasionally amplified in the lowest stringency conditions (i.e., using a reaction volume of 18 µL) and systematically amplified with a 66 °C annealing temperature. However, in both cases, all the Ct values generated were later than the Ct generated by the LOD positive control, and therefore considered as negative according to our internal rule of decision. Finally, in the context of an official analysis targeting the regulated P. citricarpa, our recommendation would be that the more specific cCBS PCR or qCBS qPCR positive results can be confirmed with the VGP qPCR test, which is a bit more sensitive but less specific, or the assay of Zajc et al. (2022). The three assays target different regions of the genome, which will secure the diagnosis.

Supplemental Information

Supplemental Information 1 The genomes Phyllosticta sp. strains used in this study.

Genomic resources included the genomes of strains sequenced in this study and 30 previously published genomes (Coetzee et al., 2021; Guarnaccia et al., 2019; Rodrigues et al., 2019) which were downloaded either from the MycoCosm database or the National Center of Biotechnology Information (NCBI) Sequence Read Archive (SRA) database. Strains highlighted in yellow correspond to those used in the phylogenetic study.

Click here for additional data file.

Supplemental Information 2 Subset of ITS, LSU and tef1 sequences that were deposited on GenBank.

Accessions in red indicate the sequences that were submitted to GenBank. Accessions in blue indicate that the isolate has the same sequence that the one submitted (indicated in red) to GenBank. Accessions in green indicate the sequences that were unique to the species, and submitted to GenBank. Accessions in black are from other studies.

Click here for additional data file.

Supplemental Information 3 Microsatellite genotyping of a subset of Phyllosticta ciricarpa and P. paracitricarpa strains.

Accessions in red indicate the sequences that were submitted to Genbank. Accessions in blue indicate that the isolate has the same sequence that the one submitted (indicated in red) to Genbank. Accessions in green indicate the sequences that were unique to the species, and submitted to Ganbank. Accessions in black are from other studies.

Click here for additional data file.

Supplemental Information 4 Assessment of repeatability and reproducibility of the qCBS real-time protocol.

Click here for additional data file.

Supplemental Information 5 Assessment of robustness of the qCBS real-time protocol.

The table reports mean Ct values generated when subtle alterations of PCR reaction volume or temperature were implemented, with the same set of templates. Parameters in bold characters correspond to the optimized conditions for the assay.

Click here for additional data file.

Supplemental Information 6 Assessment of transferability of the qCBS real-time protocol.

Different qPCR equipment and commercial qPCR master mixes were used with the same set of templates.

Click here for additional data file.

Supplemental Information 7 Genomic variations present in the P. paracitricarpa isolates in the tef1 locus.

Read alignments of the strains belonging to the P. citricarpa and P. paracitricarpa populations were checked using the Integrative Genomics Viewer (IGV).

Click here for additional data file.

Supplemental Information 8 Raw data for Table 3.

Testing of citrus fruits showing CBS-like symptoms by the cCBS PCR, qCBS qPCR, and van Gent-Pelzer et al. (2007) qPCR. The 18S uni test was conducted to verify the amplifiability of the DNA extracts. The mean and standard deviation of Ct values were generated with two replicates. N.t stands for not tested

Click here for additional data file.

The authors would like to acknowledge the INRAE MIGALE bioinformatics platform (https://migale.inra.fr/) and the Freiburg Galaxy Team from the University of Freiburg (Germany) for providing computational resources for bioinformatics. The authors would like to thank Céline Jeandel, Anses Plant Health laboratory, for the microsatellite genotyping.

Additional Information and Declarations

Competing Interests

Author Contributions

DNA Deposition

Data Availability

The authors declare there are no competing interests.

Renaud Ioos conceived and designed the experiments, performed the experiments, analyzed the data, prepared figures and/or tables, authored or reviewed drafts of the article, and approved the final draft.

Alexandra Puertolas conceived and designed the experiments, performed the experiments, analyzed the data, prepared figures and/or tables, authored or reviewed drafts of the article, and approved the final draft.

Camille Renault performed the experiments, analyzed the data, prepared figures and/or tables, authored or reviewed drafts of the article, and approved the final draft.

Aida Ndiaye performed the experiments, analyzed the data, authored or reviewed drafts of the article, and approved the final draft.

Isabelle Cerf-Wendling performed the experiments, authored or reviewed drafts of the article, and approved the final draft.

Jacqueline Hubert performed the experiments, authored or reviewed drafts of the article, and approved the final draft.

Wen Wang performed the experiments, analyzed the data, authored or reviewed drafts of the article, and approved the final draft.

Chen Jiao conceived and designed the experiments, performed the experiments, analyzed the data, prepared figures and/or tables, authored or reviewed drafts of the article, and approved the final draft.

Hongye Li performed the experiments, analyzed the data, authored or reviewed drafts of the article, and approved the final draft.

Josep Armengol performed the experiments, analyzed the data, authored or reviewed drafts of the article, and approved the final draft.

Jaime Aguayo conceived and designed the experiments, performed the experiments, analyzed the data, prepared figures and/or tables, authored or reviewed drafts of the article, and approved the final draft.

The following information was supplied regarding the deposition of DNA sequences:

The genome assemblies are available at DDBJ/ENA/GenBank: PRJNA949004.

The following information was supplied regarding data availability:

The raw measurements are available in the Supplemental Files.

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
