# Peer review of "Harnessing the power of comparative genomics to support the distinction of sister species within Phyllosticta and development of highly specific detection of Phyllosticta citricarpa causing citrus black spot by real-time PCR"

_PeerJ, doi:10.7717/peerj.16354_

## Round 0.1 · original submission · Major Revisions

Bonjour Renaud, Please, review the external reviewers' comments and suggestions. In particular please, address the major comments of reviewer #1.

Best wishes and regards
Simon Francis Shamoun
Academic Editor, PeerJ

Reviewer 1 ·

Basic reporting

The basic reporting of the manuscript is good. The authors objectives are clear and sufficient background is provided.

Experimental design

The authors attempt to tackle an important taxonomic issue that is very relevant to international phytosanitary regulations. However, there are some issues that worry me and that need to be solved before this work can be considered publishable.

In order to create a clear understanding about the issues, I’d like to briefly summarize the key objectives of the study. In short, the work is focused on developing methodology to distinguish what the authors consider to be two species: P. citricarpa and P. paracitricarpa. There is currently no reliable method available to distinguish these lineages, and it is heavily debated whether P. paracitricarpa should even be considered a separate species and not a genetic lineage of the same species. The authors intend to obtain this objective in two steps: 1) by establishing a clear phylogenetic relationship, or separation, of the supposed species and 2) by identifying genomic regions of polymorphy and by using these regions to develop diagnostic tools.

The most important issue is an error considering one of the used strains (CBS141350). In Figure 1, Table 1, and Supplementary Table 1, which are part of the phylogenetic analysis, this strain is classified as a P. citricarpa. In contrast, in Supplementary Table 2, which lists the data that provides the basis for their diagnostic tool development, this same strain is listed as P. paracitricarpa. It therefore seems that their phylogeny and the developed methods are not in agreement. For a study that is focused on distinguishing two supposed species, this is quite a serious and consequential error, with the result that either the phylogeny or the developed diagnostic tools are no longer supported.

In addition, the authors conclude that their phylogenic tree clearly separates the two supposed species. However, in order for it to be so, they had to rename one of the P. citricarpa strains to P. paracitricarpa (LSVM1238, unrelated to the abovementioned issue with CBS141350). I do not think it is good practice to rename strains based on a phylogeny, if the phylogeny itself is what is being evaluated. More crudely put: you cannot test the separation of two supposed species, find that one of the strains mixes up with the other species, and then simple rename that strain and declare that the species are properly separated.

The last important issue I would like to point out is that the genetic variation within P. citricarpa is not considered in the phylogenetic analysis. To be specific, for the development of the diagnostic tools the authors use 35 different P. citricarpa strains to make sure that the genetic variation within this species is taken into account. In contrast, for the phylogeny, they use only 8. This means that there may be much more variation within P. citricarpa than what is currently shown; there may actually be genetic lineages within P. citricarpa that are more divergent than the P. paracitricarpa lineage, and this is currently not taken into account. The variation that can be seen in de 35 different strains used for the development of diagnostic tools suggest that this may be the case. If one wants to establish a clear phylogenetic relationship, this intraspecies variation should be taken into account.

Validity of the findings

The authors claim: ‘The first achievement of this work is the confirmation that P. citricarpa and P. paracitricarpa are two distinct phylogenetic species’ (line 386-387). Taking into account the issues described under “Experimental design”, I consider this claim unsubstantiated: firstly because we are not sure of the identity of one of the strains (CBS141350). Secondly, because one strain had to be renamed (LSVM1238) in order for the species to actually be separated in the phylogenetic tree. And third, in order to determine whether two genetic lineages are part of one species or two, one needs to take into account the natural variation within the species. In my opinion, the variation within the P. citricarpa lineage has not sufficiently been taken into account in the phylogenetic analysis.

In addition, the authors show that the basis on which P. paracitricarpa is described as a separate species, specifically the polymorphisms in the tef1 gene, is unsubstantiated. One could conclude from this that perhaps P. paracitricarpa should be considered one and the same species as P. citricarpa. However, the authors do not seem to consider this, or at least do not give an explanation on why they (continue to) consider these supposed species to be two separate species instead of for instance, two genetic lineages or populations of the same species. A manuscript concerning a debated separation of two species should, at the very least, include a discussion of the aforementioned. As a result, the authors create the impression that they were determined to find a way to separate the two supposed species, rather than performing an objective study to determine whether the two supposed species should be separated.

Nonetheless, the methodology for the development of diagnostic tools and its validation are very sound and robust. If the authors were to correct the error with strain CBS141350, and were to demonstrate more robust data on the identity of strain LSVM1238 (both of which might be a substantial amount of work), they have developed an excellent method to distinguish all the strains that were included in this study. Although the presented data does not support the claim that they are able to distinguish two species, they could conclude that they are able to distinguish two lineages. However, I do worry that, as new strains are found, a new test will have to be developed. To clarify: the authors state that the previous method to distinguish P. citricarpa and P. paracitricarpa based on tef1, was developed using only two strains of P. paracitricarpa (line 89-90). This meant that previously unidentified variation in the used locus (tef1) was found very quickly, which in turn meant that a new method also had to be developed quickly. Although this is true for the paper describing the method (Zajc et al., 2022), the original description of P. paracitricarpa was based on at least six different strains of P. paracitricarpa (Guarnaccia et al., 2017) and all tef1 sequences described in this paper were taken into account in the paper by Zajc et al.. The current study used five different strains if CBS141350 and LSVM1238 are considered as P. paracitricarpa, and only three if these are not considered P. paracitricarpa. This suggests that what happened to the previous method may just as well happen to the method presented in this paper, which makes the relevance of this method questionable.

References:
Guarnaccia, V., Groenewald, J. Z., Li, H., Glienke, C., Carstens, E., Hattingh, V., Fourie, P. H., & Crous, P. W. (2017). First report of Phyllosticta citricarpa and description of two new species, P. paracapitalensis and P. paracitricarpa, from citrus in Europe. Stud Mycol, 87, 161-185. https://doi.org/10.1016/j.simyco.2017.05.003
Zajc, J., Kogej Zwitter, Z., Fišer, S., Gostinčar, C., Vicent, A., Galvañ Domenech, A., Riccioni, L., Boonham, N., Ravnikar, M., & Kogovšek, P. (2022). Highly specific qPCR and amplicon sequencing method for detection of quarantine citrus pathogen Phyllosticta citricarpa applicable for air samples. Plant Pathology. https://doi.org/10.1111/ppa.13679

Additional comments

In addition to the issues described above, there are a few smaller considerations for the authors.

- In line 47 – 48, the authors state that P. paracitricarpa has been shown to be pathogenic (Wang et al., 2023). However, the pathogenicity test that is used in this study is not validated and should be treated with caution.
- In line 116, the authors state that 15 new genomes were sequenced. In Supplemental information 1, I think I can only count 14.
- The order in Supplemental information 1 is a little confusing. It might be nice to order the strains alphabetically, or based on whether they were already sequenced.
- In Figure 2, one of the divergent regions between P. citricarpa and P. paracitricarpa is shown. In the text (line 318 – 321) the authors mention that in addition to the insertion and deletion sequences, they found several SNPs. These SNPs are a little challenging to spot in Figure 2; it would be nice if they could be highlighted.
- In Supplemental information 7, it is not possible to see which sequence belongs to which strain. It would be nice to include strain information.

Reference:
Wang, W., Xiong, T., Zeng, Y., Li, W., Jiao, C., Xu, J., & Li, H. (2023). Clonal Expansion in Multiple Phyllosticta Species Causing Citrus Black Spot or Similar Symptoms in China. J Fungi (Basel), 9(4). https://doi.org/10.3390/jof9040449

Reviewer 2 ·

Basic reporting

This manuscript compared genomics to find marker to differentiate species of Phyllosticta to develop and evaluate a qPCR for Phyllosticta citricarpa causing citrus black spot. The manuscript is very well done and have all element and credibility in it. The manuscript is clear and well written. All figures and tables are appropriated. All hypothesis is well supported by the conclusion. Great to read it.

Experimental design

The experimental design is very well done for convincing on the genome analysis for the finding of the gene regions for specificity. Moreover they develop a design of qPCR assays very well validated for sensitivity and specificity. They also have compared five different master mixes and also validated with environment sample for their detection. All information necessary are in the manuscript.

Validity of the findings

This assays and manuscript will be important for use of this regulatory fungi in Europe. With a sensitive and specific assay this will allow to discriminate with the cryptic species very difficult otherwise to differentiate. Also testing different master mixes they found that this was impacting on the sensitivity.

Additional comments

Very good work and really enjoyed reading it. Few minor elements:

1-L277 I would change the "resembled" by "look like"
2-L329-/21 and Figure 3, I this the use of "pc" is not correct and should be "pg" I would change them all.
3-Figure 3 should get the equation for the qPCR with the slope for efficiency?
4-In discussion would have been good to add if this method finding a unique region and then qPCR can be apply to other targets or stategy.

---

## Round 0.2 · accepted · Accept

Dear Dr. Ioos, Thank you for submitting a revised version of your manuscript along with the rebuttal and track changes documents. After reviewing these documents, I recommend publishing the revised version of your manuscript in PeerJ.
Best wishes
Dr. Simon Francis Shamoun
Academic Editor, PeerJ